# SMRT sequencing of the *Oryza rufipogon* genome reveals the genomic basis of rice adaptation

Wei Li[1,8], Kui Li[2,8], Ying Huang[3,8], Cong Shi[2,4,8], Wu-Shu Hu[3], Yun Zhang[2], Qun-Jie Zhang[1,2], En-Hua Xia[2], Ge-Ran Hutang[2,4], Xun-Ge Zhu[2,4], Yun-Long Liu[2], Yuan Liu[2], Yan Tong[2], Ting Zhu[2,5], Hui Huang[2], Dan Zhang[1], Yuan Zhao[6], Wen-Kai Jiang[2], Jie Yuan[3], Yong-Chao Niu[7], Cheng-Wen Gao[2] & Li-Zhi Gao [ID] [1,2✉]

Asian cultivated rice is believed to have been domesticated from a wild progenitor, *Oryza rufipogon*, offering promising sources of alleles for world rice improvement. Here we first present a high-quality chromosome-scale genome of the typical *O. rufipogon*. Comparative genomic analyses of *O. sativa* and its two wild progenitors, *O. nivara* and *O. rufipogon*, identified many dispensable genes functionally enriched in the reproductive process. We detected millions of genomic variants, of which large-effect mutations could affect agronomically relevant traits. We demonstrate how lineage-specific expansion of gene families may have contributed to the formation of reproduction isolation. We document thousands of genes with signatures of positive selection that are mainly involved in the reproduction and response to biotic- and abiotic stresses. We show that selection pressures may serve as forces to govern substantial genomic alterations that form the genetic basis of rapid evolution of mating and reproductive systems under diverse habitats.

[1] Institution of Genomics and Bioinformatics, South China Agricultural University, 510642 Guangzhou, China. [2] Plant Germplasm and Genomics Center, Germplasm Bank of Wild Species in Southwestern, Kunming Institute of Botany, Chinese Academy of Sciences, 650204 Kunming, China. [3] TGS Inc, 518000 Shenzhen, China. [4] University of the Chinese Academy of Sciences, 100039 Beijing, China. [5] College of Life Science, Liaoning Normal University, 116081 Dalian, China. [6] Yunnan Agricultural University, 650201 Kunming, China. [7] Genosys Inc, 518000 Shenzhen, China. [8] These authors contributed equally: Wei Li, Kui Li, Ying Huang, Cong Shi. ✉email: Lgaogenomics@163.com

Asian cultivated rice (*Oryza sativa* L.), which is grown worldwide and is one of the most important cereals for human nutrition, is thought to have been domesticated from an immediate ancestral progenitor, *O. rufipogon*, thousands of years ago[1–5]. During the process of domestication under intensive human cultivation, rice has undergone substantial phenotypic and physiological changes and has experienced an extensive loss of genetic diversity through successive bottlenecks and artificial selection for agronomic traits compared to its wild progenitor[6,7]. *O. rufipogon* span a broad geographical range of global pantropical regions[8], and for example, extensively occur in diverse natural habitats in South China[9,10]. Although Asian cultivated rice is predominantly selfing, estimated outcrossing rates of Asian wild rice, which ranged from ~5 to 60%, showed that mating system is associated with life-history traits and results in the differentiation into two ecotypes: predominantly selfing annual *O. nivara* having high reproductive effort and mixed-mating *O. rufipogon* with low reproductive effort[11–13]. They offer promising sources of alleles for rice improvement that is of crucial significance in world rice production and food security. Many genes involved in rice improvement have successfully been introduced through introgression lines from *O. rufipogon* and have helped expand the rice gene pool important to the generation of environmentally resilient and higher-yielding varieties[14], such as the discovery of the "wild-abortive rice" in *O. rufipogon* leading to a great success of hybrid rice[15].

Despite this great interest, assembling a typical *O. rufipogon* genome has been challenging due to the nature of outcrossing and self-incompatibility that result in a high rate of genome heterozygosity. This genomic complexity has long faced leading-edge assembly procedures compared to six other AA- genome *Oryza* species[16]. To overcome this challenge, we first present a chromosome-based assembly and annotation of the typical *O. rufipogon* genome through the integration of single-molecule sequencing, 10× and Hi–C technologies. We also performed a multi-species comparative analysis of *O. rufipogon*, *O. nivara* and *O. sativa* to offer valuable genomic resources for unlocking the untapped reservoir of this wild rice to enhance rice breeding programs.

## Results

**Genome sequencing, assembly, and annotation.** We sequenced the nuclear genome of *O. rufipogon* (RUF) from a typical natural population grown in Yuanjiang County, Yunnan Province, China. We performed a whole-genome shotgun sequencing (WGS) analysis with the single-molecule sequencing platform. This generated clean sequence data sets of ~39.47 Gb with average read length of 12.6 kb and yielded ~102.253-fold coverage (Table 1). The diploid FALCON-Unzip (version 0.3.0)[17] assembler resulted in an primary assembly of ~373.88 Mb with an contig N50 length of ~710.33 Kb (Supplementary Table 1). FALCON-Unzip also generated a combined 23.85 Mb of haplotype-resolved sequence, with an N50 of 29.47 Kb and a maximum length of 653.91 Kb (Supplementary Fig. 1; Supplementary Table 2). Both SMRT and Illumina reads were used for the correction of genome assembly. Only the corrected primary contigs were used for further scaffolding. Aided with ~39.9 Gb (~103× genome coverage) 10× data, we further assembled contigs into scaffolds with an N50 length of ~2.21 Mb (Supplementary Table 1). About 97.35% of the assembly falls into 290 scaffolds larger than 100 Kb in length (Supplementary Table 3). To obtain a chromosome-based reference genome we sequenced ~103.9 Gb (~269× genome coverage) Hi–C data and anchored ~364.46 Mb sequences into 12 pseudo-chromosomes using Lachesis[18] with default parameters based on syntenic relationship with the *O. sativa* ssp. *japonica* cv.

**Table 1 Summary of the genome assembly and annotation of *O. rufipogon*.**

| Assembly | |
| --- | --- |
| SMRT Sequencing Depth (×) | 102.3 |
| 10X Sequencing Depth (×) | 103.0 |
| Hi-C Sequencing Depth (×) | 269.0 |
| Estimated genome size (Mb) | 388.0 |
| Assembled sequence length (Mb) | 380.51 |
| Scaffold N50 (Mb) | 30.20 |
| Contig N50 (Kb) | 1096.43 |

| Annotation | |
| --- | --- |
| Number of predicted protein-coding genes | 34,830 |
| Average gene length (bp) | 2921 |
| tRNAs | 637 |
| rRNAs | 1085 |
| snoRNAs | 442 |
| snRNAs | 117 |
| miRNAs | 245 |
| Transposable elements (%) | 44.14 |

*Nipponbare* genome (MSU 7.0), representing ~94.42% of the estimated genome size of *O. rufipogon* (~386 Mb) (Supplementary Table 4). The chromosomes lengths of the RUF genome varied from ~22 Mbp (Chr12) to ~44 Mbp (Chr01) with an average size of ~30 Mbp (Fig. 1; Supplementary Fig. 2; Supplementary Table 4). The assembled genome was referred to as Oryza_rufipogon_v2.0, which showed an extensive synteny conservation with the *O. sativa* ssp. *japonica* cv. *Nipponbare* genome (MSU 7.0) (Supplementary Fig. 2). To further improve the continuity of the genome assembly, captured gaps were filled using PBJelly2[19]. Thus, we obtained an assembly of 380.51 Mb, with a contig N50 length of 1096 Kb and a scaffold N50 of 30.20 Mb (Table 1; Supplementary Table 1).

By adopting a method from Stefan et al.[20], we attempted to detect haplotype variations between primary contigs and haplotigs. The show-snp tool implemented in the MUMER package[21] was used to identity single-nucleotide polymorphisms (SNPs) and indels. After aligning the haplotigs against the genome sequence, we obtained a total of 84,227 SNPs and 54,407 indels, respectively. Using Assemblytics[22], a web-based tool, large variants (≥10 bp) between primary contigs and haplotigs were detected. A total of 704 large variants were found, including 429 insertions, 247 deletions, 9 repeat expansions, 1 repeat contractions, 16 tandem expansions, and 2 tandem contractions (Supplementary Fig. 3; Supplementary Table 5). This phased genome assembly has largely improved our understanding of haplotype composition and genomic heterozygosity within a diploid genome that will help future rice breeding efforts.

To validate the genome assembly quality, we first mapped ~33.89 Gb of high-quality reads to the assembled genome sequences, showing a good alignment with an average mapping rate of 93.0% (Supplementary Table 6); second, we aligned all available DNA, proteins of RUF from public databases and RNA sequencing (RNA-Seq) data obtained from four libraries representing major tissue types and developmental stages of the sequenced RUF individual, and obtained mapping rates of 86.94%, ~64.43%, and ~71.34%, respectively (Supplementary Table 6); and finally, we checked core gene statistics using BUSCO[23] to further verify the sensitivity of gene prediction and the completeness and appropriate haplotig merging of the genome assembly. Our gene predictions recovered 1402 of the 1440 (97.36%) highly conserved core proteins in the Embryophyta lineage (Supplementary Table 6).

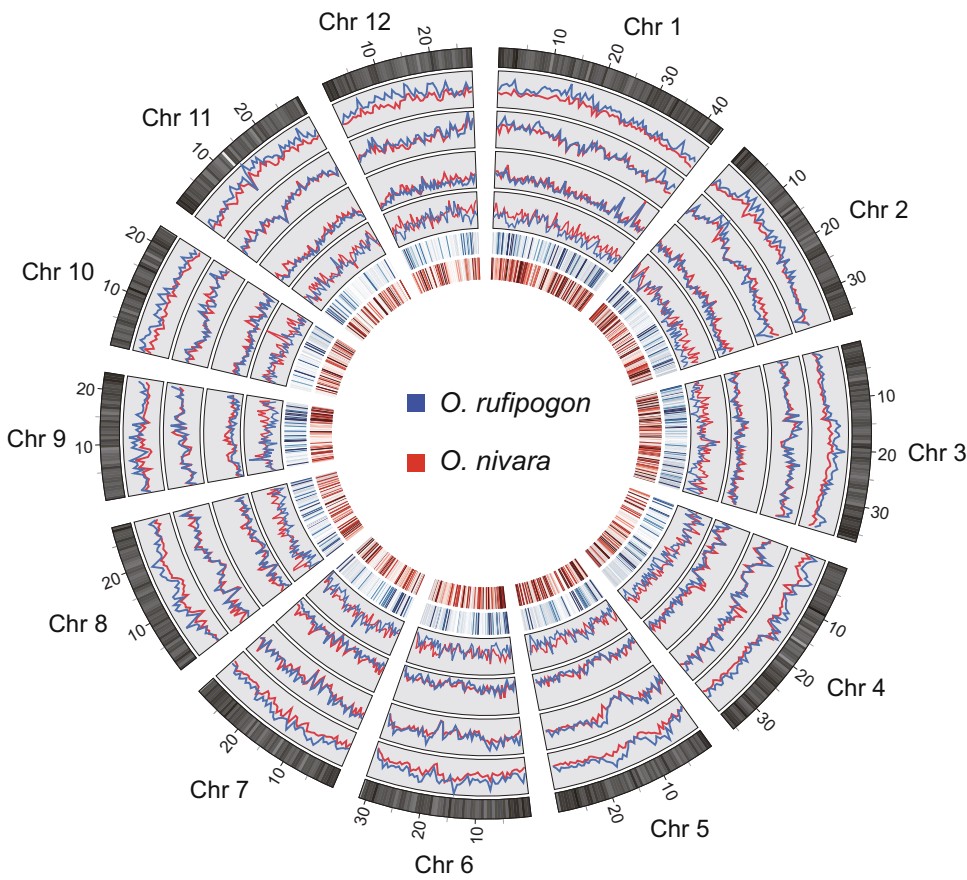

**Fig. 1 Genome feature and genomic variation of _O. rufipogon_.** The outer circle represents the 12 chromosomes of _O. sativa_, along with the gene density (non-overlapping, window size = 500 Kb). Moving inward, the four circles with line plot refer to the SNP, InDel, SV, and CNV distribution, respectively (non-overlapping, window size = 500 Kb). _O. rufipogon_ is indicated in blue, while _O. nivara_ is represented in red. The inner two circles plotted with heat map display the sequence similarities of orthologous gene pairs between _O. sativa_ and _O. rufipogon_ (blue), and between _O. sativa_ and _O. nivara_ (red).

In combination with ab initio prediction, protein and expressed sequence tags (ESTs) alignments, EvidenceModeler combing and further filtering, we predicted 34,830 protein-coding genes (Supplementary Table 7). Of them, 84.2% of the gene models were supported by transcript and/or protein evidences (Supplementary Table 8). We also annotated non-coding RNA (ncRNA) genes, including transfer RNA (tRNA) genes, ribosomal RNA (rRNA) genes, small nucleolar RNA (snoRNAs) genes, small nuclear RNA (snRNAs) genes, and microRNA (miRNAs) genes (Supplementary Table 9). In total, 245 miRNA genes belonging to 77 miRNA families were identified in the RUF genome (Supplementary Table 9). The annotation of repeat sequences showed that ~44.14% of the RUF genome consists of transposable elements (TEs), larger than the amount (39.40%) annotated in the SAT genome with the same methods (Supplementary Table 10). LTR retrotransposons were the most abundant TE type, occupying roughly 25.87% of the RUF genome. We annotated 218,967 simple sequence repeats (SSRs) that will provide valuable genetic markers to assist rice-breeding programs (Supplementary Table 11).

**Multi-species comparative analysis of and genomic variation in _O. rufipogon_, _O. nivara_, and _O. sativa_.** We performed a multi-species comparative analysis by comparing SAT with the two wild ancestral genomes, RUF and _O. nivara_ (NIV)[16] (Fig. 1; Supplementary Table 12), obtaining an overall statistic of 515,500,353 bp and a total set of 51,533 genes (Fig. 2a; Supplementary Table 13). Our results showed the increase of total genes but the reduction

of core genes from two pair rice genomes to the three genomes (Fig. 2a). The core-genome size of the three species and average pan-genome size of any two species accounted for ~61.6% (317,729,226 bp) and ~92.1% (474,815,432 bp) of whole pan-genome (Fig. 2b; Supplementary Table 13), respectively, suggesting that any single genome may not sufficiently represent the genomic diversity encompassed within the rice gene pool. Approximately 27.4% (14,135 core genes) of the protein-coding genes were conserved across all three genomes, and nearly 44.6% (22,979 genes) were present in more than one but not all three rice genomes, representing the dispensable genome. Gene Ontology (GO) enrichment analysis showed that core genes were enriched in fundamental biological processes, while the functional category of reproductive process was intriguingly enriched in dispensable genes ($P < 0.001$; FDR < 0.001) (Supplementary Table 14).

The completion of high-quality genome sequences of both cultivated _O. sativa_ and the two immediate wild progenitors, _O. rufipogon_ and _O. nivara_, enables us to detect genomic variation and characterize sequence variants of functionally important rice genes. We compared these three genomes to unearth genomic variation including single-nucleotide polymorphisms (SNPs), insertions or deletions (InDels), structural variants (SVs), copy number variation (CNVs), and presence-absence variation (PAVs) (Fig. 1; Supplementary Fig. 4). SNPs and SVs were cataloged using reads mapping analysis and the assembly-based method, yielding 4,997,466 SNPs and 817,238 InDels in RUF and 3,794,980 SNPs and 779,252 InDels in NIV as compared to Nipponbare, respectively (Supplementary Table 15). Notably,

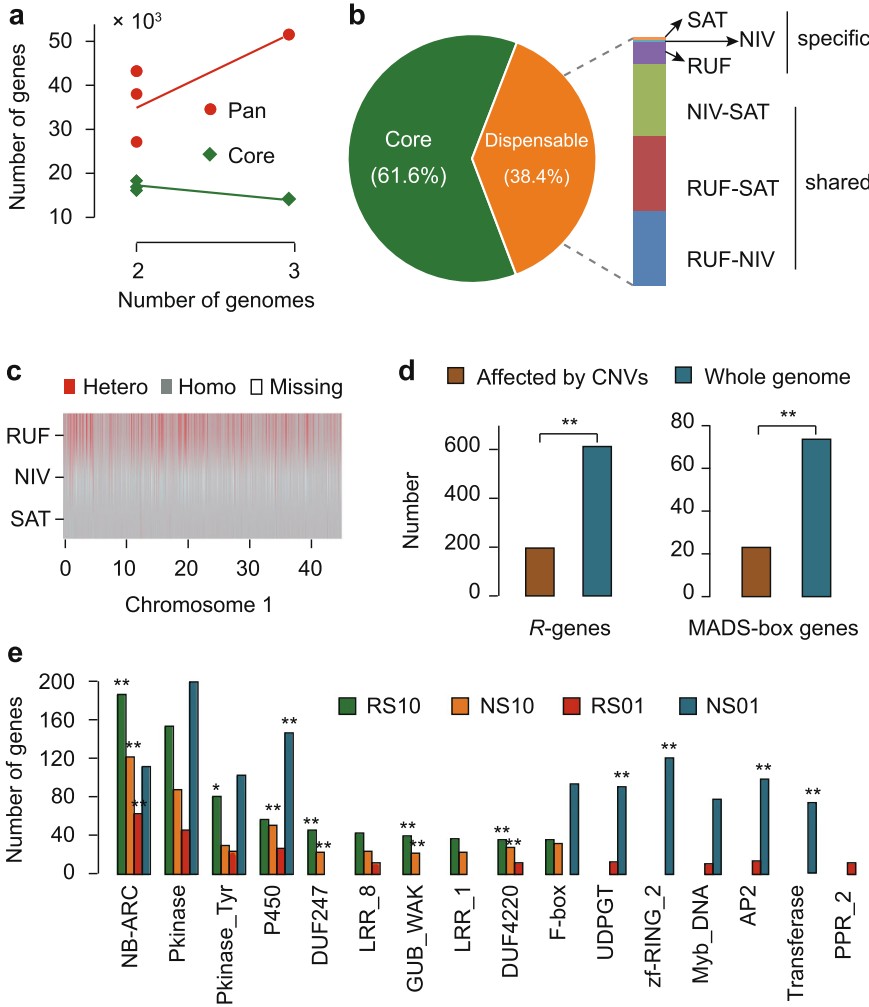

**Fig. 2 Multi-species comparative analysis and genomic variation among _O. rufipogon_, _O. nivara_ and _O. sativa_. a** Increase and decrease of gene numbers in pan- and core- genome. **b** Sequence composition of the pan-genome among RUF, NIV, and SAT. **c** Exemplar patterns of single-nucleotide polymorphisms on rice Chromosome 1 (see Supplementary Fig. 9 for the other 11 chromosomes). **d** Number of _R_-genes and MADS-box genes affected by CNVs. **e** Functional enrichment of genes affected by PAVs. The top 10 PFAM functional categories for each PAV type are shown. Asterisk indicates the significance of FDR < 0.05, while double asterisk means FDR < 0.01. PAV types are represented in a customized format, of which RS10 indicates that a PAV is present in RUF but absent in SAT, NS10 denotes that a PAV is present in NIV but absent in SAT, RS01 shows that a PAV is absent in RUF but present in SAT, and NS01 signifies that a PAV is absent in RUF but present in SAT.

both wild rice (RUF and NIV) possessed considerably larger SNPs and InDels than cultivated SAT, and the outcrossing species RUF had larger SNPs and InDels than the predominantly two selfing rice species, NIV and SAT (Supplementary Table 15). This result is in a good agreement with rather high heterozygous SNP rates throughout the RUF genome than NIV and SAT (Fig. 2c; Supplementary Fig. 5). We examined the sequence variants for their potential functional effects on protein-coding genes, and identified a total of 446,309 and 349,519 non-synonymous SNPs in RUF and NIV, respectively (Supplementary Table 16). Besides, we detected 17,124 and 14,083 SNPs that resulted in stop codon gains and 2218 and 1730 SNPs that resulted in stop codon losses in RUF and NIV, respectively (Supplementary Table 16). Although the size distribution of insertions and deletions within protein-coding sequences indicated peaks at positions that are multiples of three owing to negative selection on frame-shift InDels (Supplementary Fig. 6), 25,139 and 41,038 genomic SVs with large effect resulted in frameshifts in RUF and NIV, respectively (Supplementary Table 16). The identification of SNPs, Indels and/or SVs with large effect among SAT, RUF, and

NIV will accelerate the discovery of candidate genes related to the improvement of cultivated rice.

We integrated methods of reads mapping analysis and synteny comparisons to identify CNVs within hundreds of genes that had either gained or lost copies in RUF and NIV compared to SAT. Of 319 genes affecting both RUF and NIV, 88 had CNV loss, 145 had CNV gain, and 86 had both CNV loss and gain, while 6940 and 940 genes occurred CNV gain and loss, respectively, in either RUF or NIV alone (Supplementary Table 17; Supplementary Data 1). GO enrichment analysis indicated that genes function in flower development (GO:0009908; _P_ < 0.001) and abiotic and biotic stresses, such as stress response and resistance (_R_)-genes with nucleotide-binding site (NBS) or NBS-leucine-rich repeat (LRR) domains and transcription factors were significantly enriched in genes affected by CNVs (_P_ < 0.001) (Supplementary Data 2). Further analyses showed that a large number of genes associated with rice flower development (Supplementary Tables 18–21; Supplementary Data 3) and resistance (_R_)-genes with nucleotide-binding site (NBS) or NBS-leucine-rich repeat (LRR) domains are remarkably affected by CNVs (Fig. 2d;

Supplementary Table 22; Supplementary Data 4). The results suggest that wild rice genes affected by CNVs may be involved in flower development, flowering time, reproduction, and adaptation to changeable climatic environments and/or interaction with pathogens in nature.

Altogether, we identified 35,906 RUF-specific and 49,620 NIV–specific PAVs that account for ~26 Mbp of RUF-specific and ~32 Mbp of NIV–specific PAV (defined as >100 bp and <95% identity) (Supplementary Tables 23 and 24 and Supplementary Fig. 7). There were 7862 and 17,501 genes found to have at least 80% of their coding sequences composed of RUF- and NIV- specific sequences (Supplementary Table 24). Notably, functional annotation shows that a large number of genes affected by RUF-specific and NIV–specific PAVs are significantly enriched in functional categories involved in the disease resistance, such as NB-ARC domain (PF00931, $P < 0.001$; FDR < 0.001), Leucine rich repeat (PF13855, $P < 0.05$; FDR < 0.05), and Leucine Rich Repeat (PF00560, $P < 0.05$; FDR < 0.05), and response to environmental change, such as oxidoreductase activity (GO: 0016491, $P < 0.05$; FDR < 0.05) (Fig. 2e; Supplementary Data 5). These RUF-specific and/or NIV-specific R-genes with NBS domains, which are usually considered to mediate effector-triggered immunity acting as detectors for pathogen virulence proteins[24], represent an important portion of the dispensable rice genome, some of which possibly reflect important gene sources of wild rice for the adaptation to biotic stresses under diverse habitats.

**Accelerated evolution of rice gene families**. To examine the evolution of gene families underlying physiological and phenotypic changes we compared the predicted proteomes of RUF, NIV, and SAT, yielding a total of 29,879 orthologous gene families that comprised 100,238 genes (Supplementary Table 25). This revealed a core set of 72,490 genes belonging to 17,454 clusters that were shared among all three rice species, representing ancestral gene families in Asian cultivated rice and the two presumed wild progenitors (Fig. 3a). Interestingly, 1007 (2473 genes), 437 (1097 genes), and 239 (633 genes) gene clusters were found unique to RUF, NIV, and Asian cultivated rice (SAT) (Fig. 3a; Supplementary Table 25). Functional enrichment analyses of RUF-specific genes by both Gene Ontology (GO) terms and PFAM domains together revealed functional categories related to stress up-regulated Nod 19 (PF07712, $P < 0.001$), pathogenesis (GO:0009405, $P < 0.001$), pollen allergen (PF01357, $P < 0.001$), and root cap (PF06830, $P < 0.001$) (Supplementary Data 6). Functional enrichment analyses of NIV-specific genes showed functional categories related to petal formation-expressed (PF14476, $P < 0.001$) and photosynthesis processes, such as photosynthesis (GO:0015979, $P < 0.001$), photosystem II (GO:0009523, $P < 0.001$), photosystem II reaction center W protein (PsbW) (PF0712, $P < 0.001$) (Supplementary Data 6). Functional enrichment analyses of SAT-specific genes disclosed functional categories related to defense response (GO:0006952, $P < 0.001$), response to oxidative stress (GO:0006979, $P < 0.001$) and photosynthesis, such as photosynthesis (GO:0015979, $P < 0.001$), photosynthesis, light reaction (GO:0019684, $P < 0.001$), photosynthetic electron transport chain (GO:0009767, $P < 0.001$), photosynthetic electron transport in photosystem II (GO:0009772, $P < 0.001$), photosystem (GO:0009521, $P < 0.001$), photosystem I (GO:0009522, $P < 0.001$), photosystem II (GO:0009523, $P < 0.001$), photosystem II reaction center (GO:0009539, $P < 0.001$), photosystem II stabilization (GO:0042549, $P < 0.001$), photosystem II 10 kDa phosphoprotein (PF00737, $P < 0.001$), photosystem II 4 kDa reaction center component (PF02533, $P < 0.001$), photosystem II reaction center X protein (PsbX) (PF06596, $P < 0.001$),

photosynthetic reaction center protein (PF00124, $P < 0.001$), photosystem II protein (PF00421, $P < 0.001$) (Supplementary Data 6).

To understand the expansion or contraction of rice gene families we characterized gene families that undergo detectable changes and divergently evolve along different branches with a particular emphasis on those involved in phenotypic traits and environmental changes. Our results showed that, of the 23,755 gene families (29,193 genes) inferred to be present in the most recent common ancestor of the four studied rice species, 2486 (3567), 790 (2060), and 526 (3741) exhibited significant expansions (contractions) ($P < 0.001$; FDR < 0.001) in the RUF, SAT, and NIV lineages, respectively (Fig. 3b; Supplementary Fig. 8; Supplementary Data 7). Remarkably, functional annotation demonstrates that a large number of genes enriched in functional categories involved in the recognition of pollen (GO:0048544, $P < 0.001$) were significantly amplified in RUF but contracted in NIV in comparison with SAT (Supplementary Data 8). Compared with NIV and SAT, however, genes enriched in functional categories involved in the reproduction, including male sterility proteins (PF03015, PF07993, $P < 0.001$) and petal formation-expressed protein (PF14476, $P < 0.001$), were significantly contracted in RUF (Supplementary Data 8). Compared with RUF and NIV we surprisingly found that gene families in SAT were significantly enriched in a number of functions related to defense response (GO:0006952, $P < 0.001$), response to oxidative stress (GO:0006979, $P < 0.001$), and photosynthesis in particular, including photosynthesis (GO:0015979, $P < 0.001$), photosynthesis, light reaction (GO:0019684, $P < 0.001$), photosynthetic electron transport in photosystem II (GO:0009772, $P < 0.001$), photosystem I (GO:0009522, $P < 0.001$), and photosynthetic reaction center protein (PF00124, $P < 0.001$) (Supplementary Data 8).

Among the highly expanded and contracted gene families, we found that disease-resistance genes were significantly contracted in NIV but amplified in RUF and SAT, which are highly enriched in functional categories, including leucine rich repeats (PF12799, PF13855, PF13504; $P < 0.001$), NB-ARC domain (PF00931, $P < 0.001$), and Leucine rich repeat N-terminal domain (PF08263, $P < 0.001$) (Supplementary Data 8). Whole-genome comparative analysis of the nucleotide-binding site with leucine-rich repeat (NBS-LRR) genes further revealed a large expansion of gene families relevant to an enhanced disease resistance in RUF. In total, we identified 576, 631, and 489 genes encoding NBS-LRR proteins in RUF, SAT, and NIV, respectively (Supplementary Table 26). The contraction in NIV versus RUF is mainly attributable to a decrease in CC-NBS, CC-NBS-LRR, NBS, and NBS-LRR domains. It is noteworthy that, compared to the two wild progenitors, SAT exhibited an expansion of NBS-LRR genes, which mainly come from an increase of CC-NBS-LRR and NBS-LRR domains. We positioned these orthologous R-genes (~98%) to specific locations across the SAT chromosomes (Fig. 3c), showing an almost unequal distribution of the amplified NBS-encoding genes throughout the entire genome, particularly on Chromosome 11, which offer a large number of disease resistance candidate loci for further functional studies and rice breeding programs.

**Natural selection on rice genes**. The three fairly closely related rice genomes provide a good model to assess the adaptive evolution of rice protein-coding genes under natural selection. We identified 10,206 high-confidence 1:1 orthologous gene families that were used to construct a phylogenetic tree and estimate divergence times among RUF, NIV and SAT using *O. meridionalis* (MER) as outgroup (Supplementary Fig. 9). Average

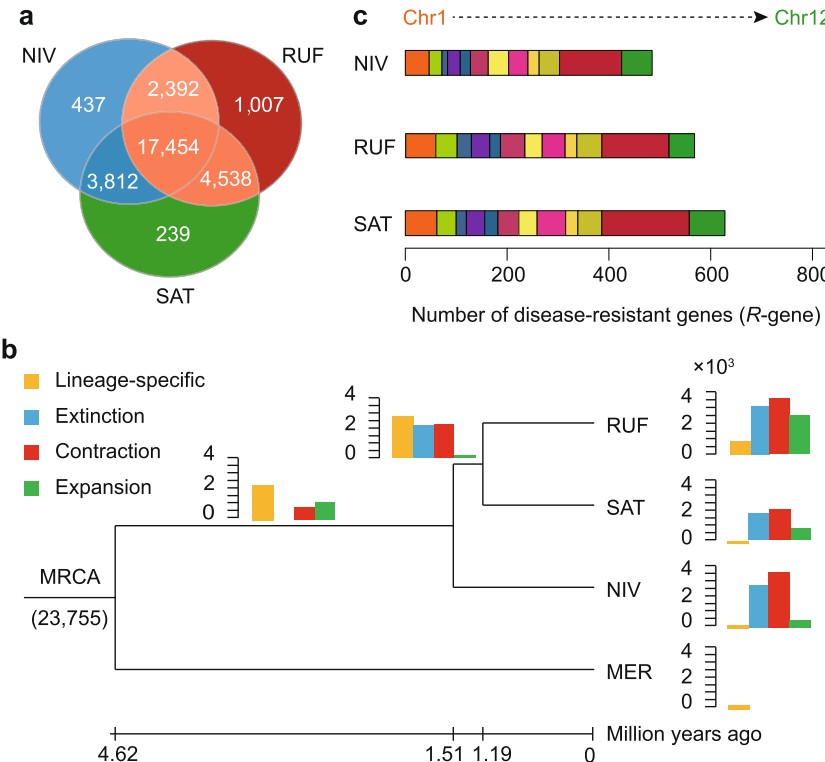

**Fig. 3 Dynamic evolution of gene families. a** Venn diagram shows the shared and unique gene families among RUF, NIV, and SAT. **b** Expansion and contraction of gene families among RUF, NIV, SAT, and MER. Phylogenetic tree was constructed based on 10,206 high-quality 1:1 single-copy orthologous genes using MER as outgroup. Bar plot beside or on each branch of the tree represents the number of gene families undergoing gain (green) or loss (red) events. Lineage-specific and -extinct families are colored in orange and light blue, respectively. Number at the tree root (23,755) denotes the total number of gene families predicted in the most recent common ancestor (MRCA). The numerical value below phylogenetic tree shows the estimated divergent time of each node (MYA; million years ago). **c** Comparisons of disease-resistant genes among RUF, NIV, and SAT.

synonymous (dS) and non-synonymous (dN) gene divergence values varied but are well comparable to the branch lengths that account for lineage divergence (Supplementary Fig. 9; Supplementary Table 27). Overall, the observed branch-specific ω values (nonsyonymous-synonymous rate ratio, dN/dS) were 0.5352, 0.6598, and 0.5382 for SAT, NIV, and RUF, respectively (Fig. 4a; Supplementary Fig. 10; Supplementary Table 27), suggesting that these three rice species may have experienced purifying selection. To test the hypothesis that the rapidly evolving genes showing increased dN/dS ratios have been under positive selection and are further promoted by speciation[25], we looked for such footprints using likelihood ratio tests for the same orthologous gene set from the three AA-genomes. Consistent with previously reported genome-wide positive selection scans in the five rice genomes[16], all tests identified a total of 2053 non-redundant positively selected genes (PSGs) (false discovery rate, FDR < 0.05) (Supplementary Table 28; Supplementary Data 9). Besides 1799 PSGs in the site model tests for all branches, we detected that a total of 90, 199, and 476 branch-specific PSGs in SAT, RUF and NIV (Fig. 4b; Supplementary Table 28). Comparing previous genome-wide scans for positive selection[26], we detected strikingly large proportions of PSGs in the overall phylogeny of rice species (~20.1%, 2,053) (Supplementary Table 28), which might be associated with the process of recent speciation and subsequently rapid adaptation to particularly varying environments.

The inclusion of the three rice genomes for all non-redundant PSGs yields a statistically significant enrichment for GO categories that span a wide range of functional categories, of which 65 genes involved in "flower development" and 51 in "response to biotic stimulus" categories showed evidence for positive selection (Fig. 4c; Supplementary Data 10). Flower

development-related traits, flowering times, the formation of reproduction, and adaptation to specific environments are crucial to and characteristic of the rapid evolution of mating and reproductive systems of these three closely related rice species inhabiting on different natural habitats. Hence, it is interesting that genes involved in flower development, reproduction, and resistance-related processes have been under positive selection in these species. With this in mind, we further examined functional enrichment for branch- or species-specific datasets of PSGs, showing that there is the largest number of PSGs in NIV (Supplementary Data 10). Notably, many candidate PSGs were significantly over-represented in categories related to ripening, flower development, pollination, reproduction and response to extracellular stimulus in NIV (P < 0.001) (Supplementary Data 10). Indeed, we detected that up to 71 genes known to play an important role in ripening (e.g., *MATE* efflux family), flower development (e.g., *OsIDS1*, *RFL*, *Hd1*, *Ehd2*, *OsSWN1*, and *OsRRMh*) and reproduction (e.g., *CSA*, *RAD51C*, *OsGAMYB*, *TDR*, *GnT1*, *DPW*, *SDS*, *OsMSH5*, *OsABCG15*, *OsCOM1*, and *OsMYB80*) pathways show signs of positive selection (Fig. 4d; Supplementary Data 11).

## Discussion

The completion of the two subspecies genomes of *O. sativa*[27–30] has greatly enhanced the identification and characterization of functionally important genes for the rice community. The availability of the first chromosome-based high-quality reference genome of *O. rufipogon*, presented here, have contiguity improvements over the published *O. rufipogon* genomes based on NGS technologies[4,31]. This typical Asian wild rice genome is, to our knowledge, the first long read assembly among numerous

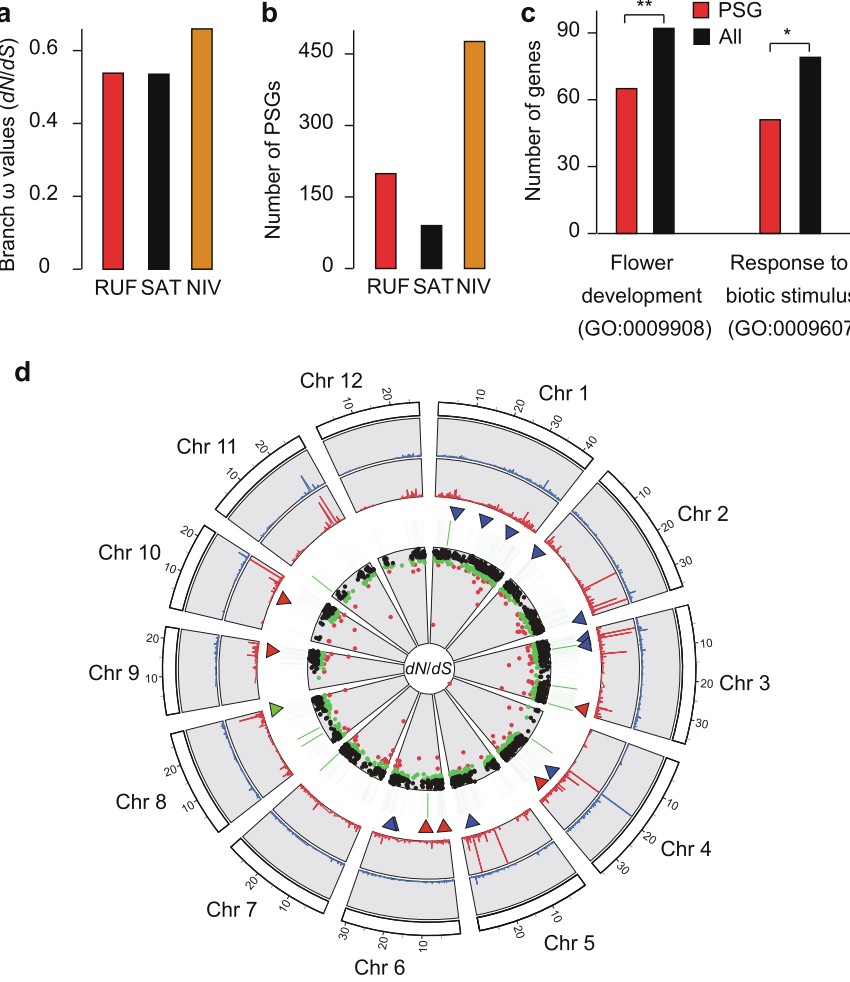

**Fig. 4 Natural selection on rice genes. a** Branch-specific ω values of RUF, SAT, and NIV estimated by using PAML. **b** Number of PSGs identified in RUF, SAT, and NIV lineage. **c** Functional enrichment of flower development and biotic stimulus response-related PSGs compared with whole gene set. **d** Genome-wide distribution of PSGs. The outer ring represents the 12 rice chromosomes; the four circles from the perimeter to the center separately refer to the dN, dS, PSGs, and dN/dS distribution for the 2053 1:1 orthologous genes. The 18 genes functionally associated with ripening (green triangles), flower development (red triangles) and reproduction (blue triangles) are marked. Black points in the inner circles show the dN/dS ratios < 0.5, while green points indicate 0.5 ≤ dN/dS < 0.8, and red points present dN/dS ≥ 0.8.

wild progenitors of domesticated crops, and now provides powerful genomic resources to investigate the orthologous loci and genomic regions associated with agronomically relevant traits of cultivated rice. The past century has witnessed the achievement to breed environmentally resilient and high-yielding rice varieties owing to the introduction of alien genes of *O. rufipogon* and other AA- genome relatives to expand the gene pool of Asian cultivated rice[32]. Thus, the completion of the *O. rufipogon* genome together with the availability of Nipponbare and six other AA-genomes[16,27,33,34] will become valuable genomic resources to enhance the exploitation of wild rice germplasms for rice genetic improvement.

Genomic variation has been extensively investigated through comparisons of genome assemblies of the *Oryza* species[16,31] and population genomic analysis of *O. rufipogon* based only on Illumina reads[35]. This study drew a map of genomic variation and addressed questions that do not largely overlap former studies[16,31,35]. We performed a multi-species comparative analysis of the annual selfing *O. sativa* and its two wild progenitors, the annual selfing *O. nivara* and perennial outcrossing *O. rufipogon*, using de novo assembly and reads mapping-based methods. This study demonstrates the advantage of multi-species comparative analysis that the cultivated rice genome alone may

not adequately represent the genomic diversity of whole rice species' gene pool. We show that a great number of dispensable genes were functionally enriched in reproductive process, possibly forming the genetic basis of a rapid evolution of mating and reproductive systems among the three rice species.

We cataloged a large data set comprising millions of genomic variants for cultivated and wild rice, of which large-effect genomic variants, including SNPs, InDels or SVs causing stop codon gain or loss and frameshift, CNV and PAVs, may affect a number of functionally important genes. These sequence variants that may associate with agronomic phenotypes or QTLs of agronomic traits will be useful in improving rice cultivars, in which rare alleles may be mined and functionally validated. They will also serve as dense molecular markers to assess new allelic combinations for marker-assisted mapping of agriculturally important traits in rice breeding programs.

Genome-wide structural variations are hypothesized to drive important phenotypic variation within a species, and a number of CNVs and PAVs in R-genes across the species have been extensively documented[36–39]. In this study, the multi-species comparative analysis showed that a large number of candidate genes affected by CNVs associate with various abiotic and biotic stresses, flower development, flowering time and reproduction.

We also captured lots of RUF-specific and NIV–specific PAVs that represent an important portion of the dispensable genome and affect genes significantly enriched in the disease resistance. Such genes and/or alleles possibly reflect important genetic materials from wild rice to adapt to diverse natural habitats, which may be exploited to enhance increased resilience to climate variability in cultivated rice.

Our analysis shows an accelerated evolution of rice gene families, a considerable portion of which were de novo generated and/or experienced fast lineage-specific expansions and contractions with significantly functional enrichment associated with physiological alterations, phenotypic variation and environmental changes from their common ancestor during the past 1.5 Myr. A large number of genes associated with the formation of reproduction isolation, such as the recognition of pollen and male sterility, were differently amplified, suggesting that the accelerated evolution of these gene families may have largely driven the variation and evolution of mating system among Asian cultivated rice and its two immediate wild progenitors. Compared with the perennial wild rice (RUF) the two annual rice species (SAT and NIV) showed a de novo generation and/or amplification of gene families significantly enriched in photosynthesis processes, possibly resulting in the observed flowering-time phenotypic variation. Our analysis showed that disease-resistance genes have been significantly contracted in NIV but amplified in SAT and RUF during the past 1.5 Myr. The expansion of this type of genes in RUF suggests that selection pressures in response to pathogenic challenge potentiated adaptations to the diverse habitats in Asia and Australia. They provide a large number of disease resistance candidate loci for further functional genomic studies and rice breeding efforts.

We identified thousands of candidate genes that may have been under positive selection in at least one of the three rice species (SAT, RUF, and NIV) during the process of speciation. Functional enrichment analysis further suggests that they are mainly involved in flower development and response to biotic- and abiotic stresses that are expected to show signatures of adaptive evolution in changeable environments. We detected the largest number of PSGs occurred in the annual wild rice (NIV) as a result of strong selection pressure, which were significantly overrepresented in functional categories related to flower development, ripening, pollination, reproduction, and response to extracellular stimulus. Our results indicate that natural selection may serve as crucial forces to drive a rapid evolution of mating and reproductive systems of these three closely related rice species inhabiting on distinctive natural habitats. Further efforts will be required to perform experiments of functional genomics to seek evidence about how these genes genetically control environmental adaptation and/or phenotypic alterations.

A large collection of genomic variation and increased knowledge of gene and genome evolution among Asian cultivated rice and its wild progenitors have made a solid foundation for searching gene sources from wild rice germplasm. The pangenome of these three rice species could be better resolved by sequencing extra rice genomes and improving individual genomes through the recent progress in SMRT sequencing technology. These advances would also enable a precise detection of small-scale structural variants as well as large-scale inversion and translocation events. Considering quick extinction and threatened status of the *O. rufipogon* populations in nature due to severe deforestation in tropical and subtropical regions[10], it is also our deep hope that the genome assembly of this wild rice species and a large data set of genomic variation will offer valuable resources to help efficient conservation of this precious wild rice species.

## Methods

**DNA and RNA extraction, library construction, and sequencing.** An individual plant of *O. rufipogon* was collected from Yuanjiang County, Yunnan Province, China. Fresh and healthy leaves were harvested and used either directly for the isolation of nuclei or immediately frozen in liquid nitrogen prior to DNA extraction. All collected samples were eventually stored at −80 °C in the laboratory after collections. High-quality genomic DNA was extracted from leaves using a modified CTAB method[40]. The quantity and quality of the extracted DNA were examined using a NanoDrop D-1000 spectrophotometer (NanoDrop Technologies, Wilmington, DE) and electrophoresis on a 0.8% agarose gel, respectively. Single-molecule long reads from the PacBio RS II platform (Pacific Biosciences, USA) were used to assist the subsequent de novo genome assembly. In brief, 20 μg of sheared DNA was used to construct three SMRT Bell libraries with an insert size of 20 kb. The libraries were then sequenced in 10 single-molecule real time DNA sequencing cells using the P6 polymerase/C4 chemistry combination, and a data collection time of 240 min per cell. A 10× Genomics library was prepared using the GemCode Instrument and sequenced on the Illumina NovaSeq platform. The Hi–C library was constructed according to a published method[41]. Nuclear DNA was cross-linked in situ, and then cut with restriction enzyme. The sticky ends of these fragments were biotinylated and then ligated to each other. After ligation, the biotinylated fragments were enriched and sheared again for the preparation of sequencing library. Finally, the library was sequenced on Illumina HiSeq X Ten platform. Besides, we constructed the four libraries for 30-d-roots, 30-d-shoots, panicles at booting stage and flag leaves at booting stage, which were sequenced on Illumina platform and de novo assembled[42].

**De novo genome assembly and quality assessment.** The assembly of PacBio long reads was performed using FALCON (version 0.3.0)[17] with the following parameters: genome_size = 380000000, seed_coverage = 30, length_cutoff_pr = 5000, max_diff = 100, max_cov = 100. This consisted of six steps involving (1) raw reads overlapping; (2) pre-assembly and error correction; (3) overlapping detection of the error-corrected reads; (4) overlap filtering; (5) constructing graph; and (6) constructing contig. These processes produced the initial contigs. The assembly was then phased using FALCON-Unzip[17] with default parameters. Two subsets of contigs were generated, including the primary contigs (p-contigs) and the haplotigs, which represent divergent haplotypes in the genome. The assemblies were aligned to the NCBI non-redundant nucleotide (nt) database to remove potential contamination from microorganisms using BLASTN. Contigs with more than 90% length similar to bacterial sequences were removed. Both p-contigs and haplotigs were polished as follows: firstly, quiver in SMRT Analysis (version 2.3.0)[43] was used for genome polishing using PacBio data with a minimum subread length = 3000 bp, minimum polymerase read quality = 0.8. Next, the Illumina data from short libraries (≤ 500 bp) were aligned to the polished assembly using BWA (version 0.7.15)[44] with default parameters, and then, Pilon (version 1.18)[45] was used for sequence assembly refinement based upon these alignments. The parameters for pilon were modified as followed: –flank 7, –K 49, and –mindepth 15. Only the primary contigs were used for further scaffolding. To link these contigs into scaffolds, 10× Genomics data were first mapped to the assembly using BWA-MEM[46], the resulting files were sorted and merged into one BAM file using samtools (version 1.9.0)[47]. The barcoding information contained in 10x linked reads was used by fragScaff[48]. The 10X Genomics scaffolds were further scaffolded using Hi–C data. Briefly, Hi–C read pairs were aligned to the scaffolds using BWA MEM algorithm. Then Lachesis[18] was used to assign the orientation and order of each sequence with the cluster number set to 12 and other parameters as default. Manual review and refinement were performed to remove the potential errors. The gaps distributed among the pseudo-chromosome were filled with the PacBio raw reads using PBJelly2[19] with parameter settings "-minMatch 8 -minPctIdentity 70 -bestn 1 -nCandidates 20 -maxScore −500 -nproc 10 –noSplitSubreads". The assembly was subject to two rounds of Pilon (version 1.18)[45] polishing to remove the sequencing errors.

Haplotype variation was detected using MUMER package (version 3.23)[21]. The error-free haplotigs were aligned to the final assembly using nucmer (version 3.23) with the parameter: -maxmatch -l 100 -c 500. The program show-snp in the MUMER package was used to identify the SNPs and indels with the options –Clr –x 1 –T. A homemade script was used to convert the output into vcf format. Variants with a length of ≤10 bp were identified as small variants. Variants larger than 10 bp were identified using Assemblytics[22].

Four approaches were used to evaluate the quality of *O. rufipogon* genome assembly. First, we mapped clean sequencing reads (~87×) from short-insert size libraries back to the assembly using BWA (version 0.7.15)[44] with default parameters. Second, All genomic and protein sequences publicly available in NCBI database (as of January, 2018) were downloaded and aligned against the genome assembly using GMAP (version 2014-10-22)[49] and genBlastA (version 1.0.1)[50], respectively. Third, RNA sequencing reads generated in this study were assembled into transcripts using Trinity (version v2.0.6)[51] with the default parameters except that the min_kmer_cov option was 2, which were then aligned back to our genome assembly using GMAP (version 2014-10-22)[49]. Finally, the completeness of the assembly was assessed with benchmarking universal single-copy orthologs (BUSCO)[23] collected from Embryophyta lineage.

**Genome annotation**. Repetitive sequences of *O. rufipogon* genome assembly were masked prior to gene prediction. A combined strategy that integrates ab initio, protein and EST evidences were adopted to predict the protein-coding genes of *O. rufipogon*. Augustus (version 3.0.3)[52], GlimmerHMM (version 3.0.3)[53] and Gen-eMarkHMM (version 3.47)[54] were used to detect the potential gene coding regions within *O. rufipogon* genome. The protein sequences from *O. sativa* ssp. *japonica* cv. Nipponbare, *O. nivara*, *O. glaberrima*, *O. barthii*, *O. glumaepatula*, *O. long-istaminata*, *O. meridionalis*, *O. brachyantha*, *Zea mays*, *Sorghum bicolor*, and *Brachypodium distachyon* were aligned to *O. rufipogon* genome assembly using GenBlastA (version 1.0.1)[50] and further refined by GeneWise (version 2.2.0)[55]. RNA-seq reads were first assembled into transcripts using Trinity (version 2.0.6)[51], and then aligned to the genome assembly using PASA (Program to Assemble Spliced Alignments)[56] to determine the potential gene structures. EVidenceModeler (EVM)[57] was used to combine all the predicted results from ab initio, protein and EST evidences into consensus gene predictions. We filtered out gene models with their peptide lengths ≤ 50 aa and/or harboring stop codons to obtain the final gene predictions of the *O. rufipogon* genome. We aligned the protein sequences of *O. sativa* ssp. *japonica* cv. Nipponbare and the RNA-seq data of *O. rufipogon* generated in this study to assess the quality of gene prediction. Putative functions of the predicted genes were assigned using InterProScan (version 5.3)[58]. PFAM domains and Gene Ontology IDs for each gene were directly retrieved from the corresponding InterPro entries.

Five types of non-coding RNA genes, including miRNA, tRNA, rRNA, snoRNA, and snRNA genes, were predicted using de novo and/or homology search methods[16]. Transposable element (TE) were annotated by integrating RepeatMasker (www.repeatmasker.org), LTR_STRUCT[59], RECON[60], and LTR_Finder[61]. Simple sequence repeat (SSR) within *O. rufipogon* genome was identified using microsatellite identification tool (MISA)[62]. The minimum numbers of SSR motifs were 12, 6, 4, 3, 3, and 3 for mono-, di-, tri-, tetra-, penta-, and hexa-nucleotides, respectively.

**Gene family clustering and evolutionary analyses**. OrthoMCL pipeline (version 2.0.9)[63] was used to identify gene families among *O. rufipogon*, *O. sativa*, and *O. nivara*. First, protein sequences of *O. sativa* and *O. nivara* were separately downloaded from MSU Rice Genome Annotation Project Database (http://rice.plantbiology.msu.edu) and *Oryza* AA Genomes Database[16]. For genes with alternative splicing, only the longest isoforms were used. Second, the filtered protein sequences from these three species were compared using all-*vs*-all Blastp with an E-value of 1E-5. Finally, the gene families among *O. rufipogon*, *O. sativa* and *O. nivara* were clustered using a Markov cluster algorithm (MCL) with an inflation parameter of 1.5.

According to the presence and absence of genes for a given species, the species-specific gene families were retrieved and classified. An update version of CAFE (version 3.1)[64] implemented with the likelihood model was used to examine the dynamic evolution of gene families (expansions/contractions). Functional enrichment analysis for genes with expansion, contraction or species-specific was performed using Fisher's exact test with false discovery rate (FDR) corrections. PFAM domains or GO terms for each gene used in functional enrichment analyses were directly extracted from the InterProScan entries.

**Phylogenetic analyses**. The orthologous and/or closely paralogous gene families among *O. rufipogon*, *O. sativa*, *O. nivara* and *O. meridionalis* were constructed using OrthoMCL pipeline (version 2.0.9)[63]. For these gene families, only those with exactly one copy within each species were retrieved and defined as conserved single-copy gene families for subsequent phylogenetic tree construction. RAxML package (version 8.1.13)[65] was used to resolve the phylogenetic relationships among these four rice species. Briefly, the coding sequences from the identified single-copy gene families were multiply aligned using MUSCLE (version 3.8.31)[66] and concatenated to a super gene sequence for phylogenetic analyses. All alignments were further trimmed using TrimAl (version 1.4)[67] with the '-nogaps' option. The JmodelTest (version 2.1.7)[68] was used to determine the best substitution models for phylogenetic reconstruction. Phylogenetic tree among *O. rufipogon*, *O. sativa*, *O. nivara*, and *O. meridionalis* was finally constructed using RAxML package (version 8.1.13)[65] based on the GTR+GAMMA model using *O. meridionalis* as an outgroup. Bootstrap support values were calculated from 1000 iterations. Divergence times among these species were estimated using the "*mcmctree*" program implemented in the PAML package[69].

**R-gene identification and classification**. Identification of *R*-genes within *O. rufipogon* genome was performed using a reiterative method[16]. Briefly, the protein sequences of *O. rufipogon* were first aligned against the raw Hidden Markov Model (HMM) of NB-ARC family (PF00931) using HMMER (version 3.1b1)[70] with default parameters. High-quality hits with an E-value of ≤1E-60 were retrieved and self-aligned using MUSCLE (version 3.8.31)[66] to construct the *O. rufipogon*-specific NBS HMMs. Based on this *O. rufipogon*-specific HMMs, scanning the whole *O. rufipogon* proteome was conducted again and genes with the *O. rufipogon*-specific PF00931 domain were defined as *R*-genes. The identified *R*-genes were further classified by TIR domain (PF01582) and LRR domain (PF00560, PF07725,

PF12799, PF13306, PF13516, PF13504, and PF13855). These two types of PFAM domains could be detected using HMMER (version 3.1b1)[70]. CC domains within *R*-genes were identified using ncoils[71] with the default parameters.

**Multi-species comparative analysis**. We performed a multi-species comparative analysis of the RUF, NIV and SAT genomes using a similar method as described in the building of the soybean pan-genome[37]. Firstly, we separately aligned the RUF and NIV genomes against the SAT genome using "*Nucmer*" program (version 3.1) implemented in the MUMmer package (version 3.23)[21] with the parameters of "-maxmatch -c 100 −l 40". We then mapped the NIV genome onto the RUF genome using MUMmer package with the parameters of "-maxmatch -c 100 -l 40". Secondly, we processed the above-generated results from whole-genome alignments (WGA) among the RUF, NIV and SAT genomes using program of "*dnadiff*" (version 1.3) implemented in the MUMmer package[21] to obtain more high-quality alignment results. Finally, we performed a tri-genome comparison among the RUF, NIV, and SAT genomes based on their pairwise WGA results using a customized perl script. The core-genome was defined as the most conserved genomic regions shared among the RUF, NIV, and SAT genomes.

**SNP and InDel identification**. Homozygous SNPs and small InDels of the RUF and NIV genomes were directly extracted from the previous one-to-one while genome alignments (WGA) using the SAT genome as a reference sequence, respectively. We then separately detected SNPs and small InDels of the RUF and NIV genomes using GATK (version 3.5)[72] based on the short read alignment results against their own genomes. We combined the results from both WGA and GATK methods to obtain the final datasets of genomic variation. We generated the SNPs and small InDels of the SAT genome based on its short reads alignment result. Putative functional effects of SNPs and InDels were annotated using the ANNOVAR package[73]. SNPs/InDels causing stop codon gain, stop codon loss and frameshift were defined as large-effect mutations.

**Copy number variation identification**. We identified the Copy number variations (CNVs) between RUF and SAT as well as NIV and SAT using CNVnator (version 0.3)[74] based on the read depth. The parameter used for CNVnator is "-call 100". The deletions/insertions with minimal length of 500 bp and read depth < 1.2 or larger than 1.8 of the mean genomic depth are deemed as candidate CNVs. A custom script was used to perform CNV annotation and genes with > 80% of its exons in CNV region are considered as candidate genes affected by CNVs.

**Presence and absence variation identification**. We characterized genomic presence and absence variation (PAV) using the same method as described in the soybean pan-genome analysis[37]. In this study, we defined and assigned four types of PAV: RS10 (presence in RUF but absence in SAT), RS01 (presence in SAT but absence in RUF), NS10 (presence in NIV but absence in SAT) and NS01 (presence in SAT but absence in NIV). To identify PAVs between the SAT and RUF genomes, we first extracted the sequences that could not be aligned to the SAT genome. We then realigned them to the SAT genome and SMRT sequences from the *indica* genome using BLAST[75], and finally filtered sequence stretches with an identity > 95%. RUF-specific sequences were obtained after excluding the potential bacterial contamination based on the BLAST alignment against NT database. Genes with > 50% CDS regions covered by RUF-specific sequences were defined as RUF-specific genes. Based on the short reads alignment results, blocks with no mapped reads by RUF were defined as SAT-specific sequences. Genomic regions with distance < 500 bp were merged into one block. Genes that overlapped these blocks with 50% length were considered as SAT-specific sequences. The same process was used to identify the PAVs between the SAT and NIV genome.

**Positively selected gene identification**. We employed the optimized branch-site model implemented in the PAML package (version 4.4)[69] to estimate the selection pressures on protein-coding genes from 1:1 high-quality orthologous gene families. We identified genes showing the positive selection in RUF, NIV and SAT lineage based on the likelihood ratio test (LTR) *P*-value. Genes with the *P*-value < 0.01 (FDR < 0.05) were retained and regarded as PSGs.

**Statistics and reproducibility**. Gene set enrichment analysis (GSEA) was performed by Fisher's Exact Test using RStudio Version 3.5.2. *P*-values < 0.05 were considered significant.

**Reporting summary**. Further information on research design is available in the Nature Research Reporting Summary linked to this article.

## Data availability

The draft genome assembly and genome annotations have been deposited in the National Genomics Data Center under the accession number PRJCA002346. Genome assembly, gene prediction, and gene functional annotations may be accessed via the web site at: www.plantkingdomgdb.com.

## Code availability

No previously unreported custom computer code or mathematical algorithm was used to generate results central to the conclusions.

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

## Acknowledgements

This work was supported by the start-up grant from South China Agricultural University, the Key Project of the Natural Science Foundation of Yunnan Province (201401PC00397) and Yunnan Innovation Team Project (2015FA030) (to L.Z.G.), Natural Science Foundation of China (31501025) (to Y.L.L.), and Natural Science Foundation of China (31601045) (to Q.J.Z.).

## Author contributions

L.Z.G. conceived and designed the study; C.S., G.R.H., X.G.Z., T.Z., D.Z., and Yuan Z. contributed to the sample preparation and genome sequencing; K.L. and W.K.J. performed genome assembly; Y.T. and H.H. performed flow cytometry experiments; W.L. and Yun Z. performed genome annotation; Y.H., E.H.X., W.S.H., Q.J.Z., Y.L.L., Y.L., Y.C.N., J.Y., and C.W.G. performed data analysis; L.Z.G. and W.L. wrote the paper; L.Z.G. revised the paper.

## Competing interests

The authors declare no competing interests.
