## [Peer Review File · Communications Biology]

Reviewers' comments:

Reviewer #1 (Remarks to the Author):

In this manuscript, Gao et al. present the first chromosome-scale genome of the ancestral rice relative *Oryza rufipogon*. This is a significant advance which will accelerate modern rice breeding programs, especially since *O. rufipogon* has already been used to introgress new alleles into the rice gene pool. The authors also perform a comparative analysis of protein coding genes from their new *O. rufipogon* genome with published *O. sativa* and *O. nivara* (another wild progenitor of cultivated rice). In general I find that the manuscript reports on an impactful development (i.e. a high quality genome of an important rice progenitor). However, I also think the authors should describe their new genome in more detail, and dial back some of the claims made in their subsequent comparative analysis (which occupies 2/3rds of the Results section). For one, they claim to have conducted a "pan-genome" analysis, however this is not strictly speaking true. A pan-genome represents the full genetic variation within a particular clade. Pan-genomes are typically produced by sequencing tens to hundreds of different varieties in a species, something the authors have not done. I strongly recommend rephrasing the relevant text as a "multi-species comparative analysis" or "cross-species genomic analysis" instead.

I also point out that a more extensive pan-genome analysis of *O. sativa* and *O. rufipogon* has already been published (Zhao et al., 2018, Nature Genetics), which used 66 different accessions (albeit using short-read sequencing). In light of this, my major recommendation is that the authors restructure their Results and Discussion to focus more strongly on characterising the high-quality genome they have created. If they do wish to make comprehensive pan-genome level conclusions, I think they would have to perform more extensive analysis with the hundreds of published accessions of *O. sativa*, *O. rufipogon* and *O. nivara*. Instead I suggest focusing on, and describing their new genome in more detail.

I list some specific suggestions and questions below:

1. Replace "pan-genome" with "comparative genomic analyses" throughout the text
2. Provide more details on the genome sequenced. For instance, the 10X genomics platform with the SMRT-sequenced, FALCON-assembled sequence should enable the researchers to obtain at least a diploid-aware assembly (and even a true diploid assembly with FALCON-Unzip), with haplotype phasing. This needs to be explained and presented in the results. The authors should present structural variation data and allele specificity between the haplotypes. This information would be valuable to inform future breeding efforts. This might require using FALCON-Unzip on your assembly, if not already done.
3. Please also present the FDR statistics along with the P-value in the main text. The FDR value is described in the Supp. Tables but only the p-value is shown in the main text.
4. In lines 345-350 the authors discuss variants detected in genes associated with important traits. It would be helpful for the reader if this information was somehow visualised in a figure rather than in the very extensive supplementary tables. For e.g. you could present the distribution of variation across different domains of the R-genes, or MADS transcription factors discussed in the text.
5. Please provide some evidence, or a reference, to support the statement in lines 370-375 that, "In comparisons to the two wild rice (RUF and NIV) we surprisingly detected that gene families in SAT were significantly enriched in a number of functions related to responses to biotic and abiotic stresses and particularly photosynthesis as well, providing evidence that SAT may harbor important gene sources response to stresses and flowering-time phenotypic adaptations to globally diverse habitats on all continents in sharp contrast to a relatively limited distribution of wild ancestors (RUF and NIV) in pan-tropical regions of Asia and/or Australia."
6. Please discuss your findings about CNVs and PAVs in R-genes between the species in more detail in

the Discussion section, and in context with other publications.

7. In Figure 2A, it is inappropriate to show a trendline for this kind of data. This figure is also not described well in the text. How was it obtained, and what does this x-axis represent, why are there multiple data-points for 2 genomes and only a single data-point for 3 genomes?

8. Please include more data/figures on the new genome. I suggest referring to classic papers on phased genome assembly (Chin et al., 2016, Nat Methods) for inspiration.

Reviewer #2 (Remarks to the Author):

The paper presents a de novo assembly of *Oryza rufipogon*, an ancestral progenitor of cultivated rice with SMRT and Hi-C data.

The authors realize a pan-genome analysis in order to compare this progenitor with *O. sativa* and its another progenitor *O. nivara*.

Comparative genomic analyses show important mutations that may affect agronomically significant traits. This new genome, the first chromosome-based of rice, paves the way for rice design.

I am very supportive of the goal of this paper. It is very important to obtain high quality genomes for crop design.

Nevertheless, the lack of details in the methodology part does not permit to be completely confident in the results. You can find details and recommendations below in the Major recommendations section.

Without additional explanations and/or details, I recommend a major revision of this paper before the acceptance in *Communications Biology*.

Reviewer #3 (Remarks to the Author):

Gao et al. presented a draft genome of *O. rufipogon* (RUF) and some comparative analyses with several *Oryza* species in this ms. I would say that sequencing the RUF genome itself is very important and valuable for the rice research community. However, their results are not convincing to make me believe that the assembly is of high quality.

First, the assembly contiguity (with contig N50 of only 710 kb) is not good at all for nowadays.

Secondly, it was known that the *O. rufipogon* genome had the nature of high heterozygosity. Hence, just like the authors stated, assembling its genome became an "extremely challenging" task. In this study, I didn't see how they removed the obstacle and rebuilt two haplotypes that completely represented the real state of the RUF genome. Basically, a primary-contigs-based genome (only a "mixture" haploid) is not enough to be considered as a reference for this species. Too much allelic difference information may be overridden, as analysis methods and tools used in ms were unable to detect those differences and adjust algorithms automatically, which must cause many misleading results in downstream analyses.

It will be greatly appreciated if the authors could make contributions on providing a high-quality diploid genome assembly to the science community rather than just publish a draft one for the purpose of only a paper. Plus, comparative analysis of two haplotypes within the RUF genome might provide another angle to reveal the genomic basis of rice adaptation.

Reviewer #4 (Remarks to the Author):

This manuscript from Li-Zhi Gao et al. describes the sequencing of *Oryzae rufipogon* and includes a variety of comparative analyses to other rice cultivars and species.

I think that overall the content of the manuscript is solid, the authors perform a variety of analyses and generally find some interesting results. It certainly fits within the scope of *Communications Biology* as a journal.

However, I also found some portions of the manuscript quite difficult to parse in terms of how the analyses and assembly were performed and there were numerous places where evolutionary inferences were overinterpreted. Taken all together these things really dampened my enthusiasm for the manuscript.

For instance, it is very unclear from the methods section how this genome was assembled. Yes there are long reads from a few SMRT cells and these are assembled by HGAP, but I'm left unsure of how the 10x reads and Hi-C reads were incorporated. In the results section, these are split into a few different sentences that progress sequentially (L78-L88) but why not just report the final assembly metrics and describe how this was performed in the methods section?

I also find that I can't accurately evaluate the assembly because the reads are not available (it says that they will become available upon acceptance, but this makes it impossible to evaluate things as a reviewer. Plus, there's no assurance that the reads will eventually be made available).

Lastly, while the evolutionary analyses are generally pretty interesting, the authors need to rewrite these sections to relay the information more clearly. In many cases the authors point out values or ω for the genomes (L280-281) of around 0.5-0.6 and say that this is evidence of "strong purifying selection" when given all the assumptions of dN/dS analyses it's really difficult to understand what's going on at all in terms of these numbers and selection across the genomes. Likewise, there is much talk about speciation and positive selection, but without much discussion about how much introgression has occurred between the lines (earlier the authors make the case that introgression from *O. rufipogon* provides domestic rice with potentially numerous traits) but these same introgressions will affect any interpretations of selection and orthology.

Overall I find the analyses interesting, but the way they are presented is quite confusing. I would suggest that the authors rewrite and focus on clarity of presentation, and then provide a much better context for what the evolutionary analyses say and how these can be interpreted in terms of natural and domesticated populations.

Reviewers' comments:

Reviewer #1 (Remarks to the Author):

In this manuscript, Gao et al. present the first chromosome-scale genome of the ancestral rice relative *Oryza rufipogon*. This is a significant advance which will accelerate modern rice breeding programs, especially since *O. rufipogon* has already been used to introgress new alleles into the rice gene pool. The authors also perform a comparative analysis of protein coding genes from their new *O. rufipogon* genome with published *O. sativa* and *O. nivara* (another wild progenitor of cultivated rice). In general I find that the manuscript reports on an impactful development (i.e. a high quality genome of an important rice progenitor). However, I also think the authors should describe their new genome in more detail, and dial back some of the claims made in their subsequent comparative analysis (which occupies 2/3rds of the Results section). For one, they claim to have conducted a "pan-genome" analysis, however this is not strictly speaking true. A pan-genome represents the full genetic variation within a particular clade. Pan-genomes are typically produced by sequencing tens to hundreds of different varieties in a species, something the authors have not done. I strongly recommend rephrasing the relevant text as a "multi-species comparative analysis" or "cross-species genomic analysis" instead.

Response: We would highly appreciate this reviewer's encouragement and valuable comments. In the revised manuscript, we described the genome in detail and carefully improved the Result section with the emphasis on the statement about subsequent comparative genomic analysis. We agree with you that pan-genomes are typically produced by sequencing tens to hundreds of different varieties in a species. Thus, we rephrased the "pan-genome" analysis as "multi-species comparative analysis" in the revised manuscript.

I also point out that a more extensive pan-genome analysis of *O. sativa* and *O. rufipogon* has already been published (Zhao et al., 2018, Nature Genetics), which used 66 different accessions (albeit using short-read sequencing). In light of this, my major recommendation is that the authors restructure their Results and Discussion to focus more strongly on characterising the high-quality genome they have created. If they do wish to make comprehensive pan-genome level conclusions, I think they would have to perform more extensive analysis with the hundreds of published accessions of *O. sativa*, *O. rufipogon* and *O. nivara*. Instead I suggest focusing on, and describing their new genome in more detail.

Response: We would highly appreciate this reviewer's very valuable comments. We agree with you that there was a more extensive pan-genome analysis of *O. sativa* and *O. rufipogon* which has been published. Therefore, we seriously took the reviewer's recommendation, and we reorganized and revised Results and Discussion with an emphasis on the description of this newly obtained high-quality genome assembly.

I list some specific suggestions and questions below:

1. Replace "pan-genome" with "comparative genomic analyses" throughout the text

Response: Many thanks again for the valuable suggestion. Done!

2. Provide more details on the genome sequenced. For instance, the 10X genomics platform with the SMRT-sequenced, FALCON-assembled sequence should enable the researchers to obtain at least a diploid-aware assembly (and even a true diploid assembly with FALCON-Unzip), with haplotype

phasing. This needs to be explained and presented in the results. The authors should present structural variation data and allele specificity between the haplotypes. This information would be valuable to inform future breeding efforts. This might require using FALCON-Unzip on your assembly, if not already done.

Response: Many thanks for valuable comments. In the revised manuscript, we provided detailed explanation about how we combined the 10X genomics platform with the SMRT-based sequencing technology to obtain the high-quality genome assembly; we also used FALCON-Unzip to generate phase genomes and characterize structural variation and allele specificity between the haplotypes.

3. Please also present the FDR statistics along with the P-value in the main text. The FDR value is described in the Supp. Tables but only the p-value is shown in the main text.

Response: We would highly appreciate valuable suggestions. Done!

4. In lines 345-350 the authors discuss variants detected in genes associated with important traits. It would be helpful for the reader if this information was somehow visualised in a figure rather than in the very extensive supplementary tables. For e.g. you could present the distribution of variation across different domains of the R-genes, or MADS transcription factors discussed in the text.

Response: We would highly appreciate valuable suggestions. We think that, for the readers in the rice research community, such detailed information presented in very extensive supplementary tables may be sufficiently useful for further functional genomic studies, which are urgently needed. However, we agree with you that we present the distribution of variation across different domains of the R-genes or MADS transcription factors should be helpful to further understand patterns of variation existed across these rice species. Note that we have been working on hundreds of high-quality cultivated and wild rice genomes using SMRT sequencing platform, and we are sure to take this idea in subsequent data analyses.

5. Please provide some evidence, or a reference, to support the statement in lines 370-375 that, "In comparisons to the two wild rice (RUF and NIV) we surprisingly detected that gene families in SAT were significantly enriched in a number of functions related to responses to biotic and abiotic stresses and particularly photosynthesis as well, providing evidence that SAT may harbor important gene sources response to stresses and flowering-time phenotypic adaptations to globally diverse habitats on all continents in sharp contrast to a relatively limited distribution of wild ancestors (RUF and NIV) in pan-tropical regions of Asia and/or Australia."

Response: We would highly appreciate valuable suggestions. We carefully checked our obtained results, extensively searched for relevant references and seriously examined the context of the discussion. It seems that this inference may be overstated and thus we decided to remove it in the manuscript.

6. Please discuss your findings about CNVs and PAVs in R-genes between the species in more detail in the Discussion section, and in context with other publications.

Response: We would highly appreciate valuable suggestions. Done!

7. In Figure 2A, it is inappropriate to show a trendline for this kind of data. This figure is also not described well in the text. How was it obtained, and what does this x-axis represent, why are there multiple data-points for 2 genomes and only a single data-point for 3 genomes?

Response: We would highly appreciate the reviewer's valuable suggestions. Following the first pan-genome analysis in soybean, there are a number of studies showing that fewer core genes but more pan genes while additional genomes are included. Thus, we think that it should be fine to present this kind of data in this study. In the revised manuscript, we have well described Figure 2A in the text. We would clarify that this figure indicates how core genes vary in terms of the compared genomes. When any two pairs of the three genomes (sat, niv and ruf) are compared, multiple data-points are used to show core and pan genes in terms of pairwise genomes (sat-niv, sat-ruf and niv-ruf); in contrast, when all three genomes (sat, niv and ruf) are compared, there is only a single data-point for the number of core and pan genes, respectively. We hope that we have clearly explained the obtained results in this figure.

8. Please include more data/figures on the new genome. I suggest referring to classic papers on phased genome assembly (Chin et al., 2016, Nat Methods) for inspiration.

Response: We would highly appreciate valuable suggestions. By following this reviewer's valuable suggestion, we added Figures S1 and 3 and Tables S2 and S5 in the revised manuscript.

Reviewer #2 (Remarks to the Author):

The paper present a de novo assembly of *Oryza rufipogon*, an ancestral progenitor of cultivated rice with SMRT and Hi-C data. The authors realize a pan-genome analysis in order to compare this progenitor with *O. sativa* and its another progenitor *O. nivara*. Comparative genomic analyses show important mutations who may affect agronomically significant traits. This new genome, the first chromosome-based of rice, pave the way for rice design. I am very supportive of the goal of this paper. It is very important to obtain high quality genomes for crop design. Nevertheless, the lack of details in methodology part do not permit to be completely confident in the results. You can find details and recommendations below in the Major recommendations section. Without additional explanations and/or details, i recommend a major revision of this paper before the acceptance in Communications Biology.

Response: We would highly appreciate this reviewer's encouragement and valuable suggestions. In the revised manuscript, we carefully improved the methodology by providing detailed information and/or explanation. We believe that we have clearly described the obtained results and in-depth discussion in the revised version of manuscript.

Major recommendations. The results presented in this study are closely linked to my concerns about methodology. I will not discuss in this review the results without additional informations about the methods. The authors should provide the exact set of options for each step, so that the pipeline is reproducible.

Response: We would highly appreciate valuable suggestions. We totally understand your concerns, and thus, we have provided additional information, that is, the exact set of options for each step in the revised manuscript.

1. The assembly was performed with Falcon.

The very first step of the pipeline is the assembly and it is unclear for me what has been done with Falcon. It is impossible to redo the assembly. The authors did not provide the set of options selected. Quiver was used on the output of Falcon, but on which files precisely. Do you merged p_ctg and a_ctg files before the Quiver step ? Did you ran Falcon-unzip ?

Response: We would highly appreciate these valuable suggestions. Note that we have offered detailed information of the genome assembly process in the revised manuscript. We would further explain as below: *De novo* assembly of long reads was first performed using FALCON. After the initial assembly, FALCON-Unzip was then used to produce primary contigs (p-contigs) and associate contigs (a-contig). Both the p-contigs and a-contigs were polished using Quiver, respectively.

The bioinformaticians know that the assemblies are never fully correct. So, it is important to test different options and different assemblers. It is not mention in the paper that tests were carried out. Authors can for example try CANU, wtdbg2, flye, Falcon (with different thresholds for repeats,

reads quality, reads lengths, overlap lengths). The comparison in terms of N50, size, repeats %, contigs number must be presented and discuss in the paper. By reading the few lines about the assembly, it give the impression that Falcon was run with default parameters without test. It is very important, especially since *O.rufipogon* has a high rate of heterozygosity.

Response: We would highly appreciate valuable comments. We agree with you that it is essential to test different assemblers such as CANU, wtdbg2, flye and Falcon. However, our lab has long assembling a number of large and complex plant genomes such as tea tree. We did test almost all these assemblers and found out that, like other labs, Falcon absolutely works best to assemble the heterozygous genomes with SMRT long reads. To avoid problems due to genomic heterozygosity, in this study, we selected FALCON and FALCON-Unzip as the assemblers, which could provide the best separation of haplotype blocks as well as the best contiguity. Please note that the used parameters were modified based on the characteristics of SMRT reads and genomic features as well. We have provided the detailed parameters in the revised manuscript.

Finally, The authors have Hi-C data and SMRT data, this allow to completely phase the genome. It would produce very important data for crop design. It would be interesting to see the genomic variations after Falcon-phase. The choice to not run Falcon-phase must be explained.

Response: We would really appreciate these valuable suggestions. To solve the problem due to high levels of genomic heterozygosity, we selected FALCON and FALCON-Unzip as the assemblers and analyzed genomic variation between the two contig datasets, which should be somehow sufficient in this study. However, we fully agree with you that Falcon-phase should work great to examine the genomic variations after Falcon-phase. Thus, we are using Falcon-phase to detect genomic variations in our on-going pan genome project of this wild rice species.

2. Quiver - BWA - Pilon

For each step, please mention exactly the options and/or thresholds. Using illumina polishing can be deleterious to the assembly, see my comment on quality evaluation.

Response: We would really appreciate these valuable comments. Please note that we have given detailed options and/or thresholds in the revised manuscript.

3. Hi-C data

The authors use Lachesis to assemble the scaffolds into chromosomes. Ghurye et al., 2017 (<https://bmcbgenomics.biomedcentral.com/articles/10.1186/s12864-017-3879-z#Sec8>) present SALSA with an extensive comparison to LACHESIS. SALSA present significant improvement

over LACHESIS. SALSA is not mention in the present study. The authors should justify their choice to use LACHESIS, for example by testing SALSA on their dataset.

Response: We would highly appreciate valuable suggestions. We have tested our dataset using SALSA2. Our result showed that SALSA2 assigned contigs into multiple scaffolds instead of one pseudo-chromosome. In fact, Zhang et al., 2019 (<https://www.nature.com/articles/s41477-019-0487-8>) also reported a similar result for the rice Nipponbare genome.

4. Quality evaluation

I appreciate the efforts of the authors for the quality evaluation of the assembly. The values of mapping rates presented in the results are not discussed in the paper. The reader have no idea if 86.94 % of DNA mapping or if 64.43 % of proteins mapping is good or not. This values should be compare with similar values on other SMRT genomes. I also suggest to use this pipeline (<https://www.nature.com/articles/s41587-018-0004-z>) of Watson and Warr to evaluate assembly/prediction quality (Link to point 2.).

Response: We would highly appreciate valuable suggestions. The mapping rates of short reads and proteins were relatively low which may come from high level of genomic heterozygosity of this wild rice species. In fact, we used strict criteria at 90% coverage and 90% identity thresholds may result in relatively low mapping rates. Watson and Warr proposed the pipeline which mainly focus on the impact of indel errors in long-reads assemblies by performing transcripts alignments. In this study, we have performed transcripts mapping to evaluate the assembly quality using GMAP.

The quality evaluation is not complete without a contamination analysis. This step should be mandatory to publish a genome. This can be done easily using Kraken from Salzberg lab (or Clark). The authors can create a new Kraken Database with bacteria from ensembl (after Checkm <http://ecogenomics.github.io/CheckM/> and Drep <https://github.com/MrOlm/drep>), Human samples, Fungi and Plant genomes. The selected assembly can be pass into Kraken in order to provide contamination values.

Response: We would highly appreciate valuable suggestions. We have removed contigs with more than 90% of length similar to bacterial or virus sequences. Thus, it seems not necessary to perform contamination analysis in the quality evaluation section.

5. Comparison with other genomes.

This study present extensive comparisons with the genomes of *O.sativa* and *O.nivara*. It would be very helpful to have informations about the quality of this genomes in a main table: the sequencing technology (illumina), the completeness (BUSCO), N50, Total base, number of proteins.

Response: We would highly appreciate valuable suggestions. By following this reviewer's valuable suggestion, we added Tables S12 in the revised manuscript.

Minor recommendations. 6. Phylogenetic analysis

I agree that the selection of OG with one copy allow the authors to run at the standard inflation of 1.5 but their should avoid the term "closely paralogous". The authors can justify the choice of OrthoMCL. OrthoFinder

(<https://genomebiology.biomedcentral.com/articles/10.1186/s13059-015-0721-2>) is frequently used and show improvement over OrthoMCL. This could really be an issue for larger phylogeny, but i guess that here orthoMCL can be ok. Nevertheless, the supermatrix should at least be treated with BMGE to select unambiguously aligned position.

Response: We would highly appreciate valuable suggestions. We totally agree with you that OrthoFinder can make better performance over OrthoMCL in some cases, and thus, we will try this methodology in our future genome projects. Indeed, there are several software packages aimed at detecting unreliable alignment columns, such as Gblocks, Trimal, Noisy, Alisco and BMGE. In this study, we used Trimal to remove poorly aligned genomic regions. Note that detailed options have been provided in the revised manuscript.

My recommendations are only done in order to improve the quality of the paper. I know it take energy/money to redo some analysis. I have some concerns about the methods but with these recommendations, the paper can be accepted for publication. I wish good work to the authors.

Response: We would highly appreciate this reviewer's encouragement and valuable suggestions. In the revised manuscript, we paid attention to solving these problems and we deeply believe that the paper qualifies for publication.

Reviewer #3 (Remarks to the Author):

Gao et al. presented a draft genome of *O. rufipogon* (RUF) and some comparative analyses with several *Oryza* species in this ms. I would say that sequencing RUF genome itself is very important and valuable for rice research community. However, their results are not convincing to make me believe that the assembly is of high quality. First, the assembly contiguity (with contig N50 of only 710 kb) is not good at all for nowadays. Secondly, it was known that the *O. rufipogon* genome had the nature of high heterozygosity. Hence, just like the authors stated, assembling its genome became an “extremely challenging” task. In this study, I didn’t see how they removed the obstacle and rebuilt two haplotypes that completely represented the real state of the RUF genome. Basically, a primary-contigs-based genome (only a “mixture” haploid) is not enough to be considered as a reference for this species. Too much allele difference information may be overridden, as analysis methods and tools used in ms were unable to detect those differences and adjust algorithms automatically, which must cause many misleading results in downstream analyses. It will be greatly appreciated if the authors could make contributions on providing a high-quality diploid genome assembly to the science community rather than just publish a draft one on purpose of only a paper. Plus, comparative analysis of two haplotypes within the RUF genome might provide another angle to reveal the genomic basis of rice adaption.

Response: We would highly appreciate this reviewer’s encouragement and valuable comments. We were also not satisfied with the the assembly contiguity (with contig N50 of ~710 kb only) due to the high genomic heterozygosity—note that we always obtain much better results for other plant genomes in our lab. By following this reviewer’s valuable suggestion, we re-assembled the genome and obtained an improved genome assembly of 380.51 Mb, with a contig N50 length of 1,096 Kb and a scaffold N50 of 30.20 Mb. We also rebuilt the two haplotype genomes and detected haplotype variations between primary contigs and haplotigs.

Reviewer #4 (Remarks to the Author):

This manuscript from Li-Zhi Gao et al. describes the sequencing of *Oryzae rufipogon* and includes a variety of comparative analyses to other rice cultivars and species. I think that overall the content of the manuscript is solid, the authors perform a variety of analyses and generally find some interesting results. It certainly fits within the scope of *Communications Biology* as a journal. However, I also found some portions of the manuscript quite difficult to parse in terms of how the analyses and assembly were performed and there were numerous places where evolutionary inferences were overinterpreted. Taken all together these things really dampened my enthusiasm for the manuscript. For instance, it is very unclear from the methods section how this genome was assembled. Yes there are long reads from a few SMRT cells and these are assembled by HGAP, but I'm left unsure of how the 10x reads and Hi-C reads were incorporated. In the results section, these are split into a few different sentences that progress sequentially (L78-L88) but why not just report the final assembly metrics and describe how this was performed in the methods section?

Response: We would highly appreciate this reviewer's encouragement and valuable comments. In the revised manuscript, we have seriously incorporated his/her helpful suggestions to improve the manuscript. For example, we gave detailed information about how genome assembly and data analyses were performed. We also paid particular attention to revise the discussion such as evolutionary inferences which might be over interpreted. Please kindly note that we have carefully revised the method section and clearly reported how we assembled this high-quality genome by combining SMRT, 10x and Hi-C reads.

I also find that I can't accurately evaluate the assembly because the reads are not available (it says that they will become available upon acceptance, but this makes it impossible to evaluate things as a reviewer. Plus, there's no assurance that the reads will eventually be made available).

Response: We would highly appreciate this reviewer's valuable comments. In the submitted manuscript, we state that, as usual, the reads will become available upon acceptance of the manuscript. Like all previously published genome papers, we were sure to make all reads eventually publicly available. Also, it is always required to submit reads data to the genebank by providing accession numbers. We understand that this may make it impossible to evaluate as a reviewer. In the revised manuscript, we kept working on the improvement of genome assembly, and we believe that the quality of genome assembly accuracy should be guaranteed for nowadays. Again, we would clarify that we have carefully revised the method section and clearly reported how we assembled this high-quality genome by combining SMRT, 10x and Hi-C reads. If this reviewer still has concern about the quality of genome assembly, please kindly let us know and we are more than happy to provide data sets for the review purpose.

Lastly, while the evolutionary analyses are generally pretty interesting, the authors need to rewrite these sections to relay the information more clearly. In many cases the authors point out values or omega for the genomes (L280-281) of around 0.5-0.6 and say that this is evidence of "strong purifying selection" when given all the assumptions of dN/dS analyses it's really difficult to understand what's going on at all in terms of these numbers and selection across the genomes. Likewise, there is much talk about speciation and positive selection, but without much discussion about how much introgression has occurred between the lines (earlier the authors make the case that

introgression from *O. rufipogon* provides domestic rice with potentially numerous traits) but these same introgressions will affect any interpretations of selection and orthology.

Response: We would highly appreciate this reviewer's valuable comments. We have seriously revised the completed evolutionary analyses as well as corresponding discussion, particularly by following the reviewer's fairly helpful suggestions. We hope that we have made clear and reasonable statements in the revised manuscript.

We would explain that we indeed stated that “Many alien genes involved in rice improvement have successfully been introduced through introgression lines from *O. rufipogon* and have helped expand the rice gene pool important to the generation of environmentally resilient and higher-yielding varieties”. However, please note that the introgression we mentioned here was artificially made instead of natural introgression. We understand that these same introgressions will affect any interpretations of selection and orthology. This is why we selected to sequence the individual plant of *O. rufipogon* from a typical natural population grown in Yuanjiang County, Yunnan Province, China, to maximally, if not all, prevent from historically genomic introgression, which may affect data interpretation.

Overall I find the analyses interesting, but the way they are presented is quite confusing. I would suggest that the authors rewrite and focus on clarity of presentation, and then provide a much better context for what the evolutionary analyses say and how these can be interpreted in terms of natural and domesticated populations.

Response: We would highly appreciate this reviewer's valuable comments. In the revised manuscript, we attempted to revise results and the discussion with emphasis on evolutionary analyses. We would further explain that, in this study, we examined natural selection across cultivated rice and its two wild progenitors, which have evolved together in the agroecosystem. Thus, we think that it may be reasonable to interpret data all together instead of clearly distinguishing domesticated and wild rice species.

REVIEWERS' COMMENTS:

Reviewer #1 (Remarks to the Author):

I thank the authors for addressing all my comments satisfactorily, including performing further analysis, and recommend acceptance of this manuscript.

Reviewer #2 (Remarks to the Author):

The paper present a de novo assembly of *Oryza rufipogon*, an ancestral progenitor of cultivated rice with SMRT and Hi-C data. The authors realize a multi-species comparative analysis in order to compare this progenitor with *O.sativa* and its another progenitor *O. nivara*. Comparative genomic analyses show important mutations who may affect agronomically significant traits. This new genome is the first chromosome-based of rice.

I found the paper well written and very well described in terms of methodology. I congratulate the authors for the improvements. I am convinced that the methodological choices are well thought out which was not the case in the first version. The haplotypes resolution greatly improve the manuscript. I found very easy to inspect methodology.

The authors answer to all my methodological doubts in a very detailed way. It is an important paper for the field and I recommend this paper for publication.

Reviewer #4 (Remarks to the Author):

The manuscript from Li et al describes the sequencing and some genomic comparisons of the *Oryza rufipogon* genome. This manuscript is significantly improved from the prior version. My main remaining critique (which is highlighted in a couple of comments below, but which applies to the entirety of the manuscript in a variety of places, is that the authors often couch their language in the context of investigating "adaptation" and "diversification", etc...

However, the authors have done a lot of work to identify changes and differences across these genomes but simply identifying the changes doesn't necessarily mean that they are involved in adaptation or are causative of diversification of phenotypes. The authors should rephrase the manuscript throughout to deemphasize adaptation and diversification and to simply focus on identifying changes across genomes without having to attribute these to any evolutionary process.

L26: would change the word "massive" here to "many"

L28: the authors don't really follow up on large effect mutations here, so best to just say that many agriculturally relevant traits could be affected?

L32: better as "genes with signatures of positive selection"

L36: delete "powerfully"

L54: delete "alien"

L60: delete "extremely"

L214: you don't really investigate adaptation here, I'd change the title to reflect the analyses better

L243-248: can delete these, it's speculation that doesn't add much to the paper

L249-250: you don't have the data to link these changes to phenotypic diversification. Would be best to just say you're investigating expansion and contraction of rice families

L300: you can't really read anything into overall numbers of ~ 0.5 for dN/dS . There's a lot that can go into that. In any event, it doesn't scream purifying selection

L345: again, you don't really investigate adaptation. You identify changes, but it's hard to link these to any given adaptive event

Reviewers' comments:

Reviewer #1 (Remarks to the Author):

I thank the authors for addressing all my comments satisfactorily, including performing further analysis, and recommend acceptance of this manuscript.

Response: We would again highly appreciate this reviewer's encouragement and fairly valuable comments in the last evaluation report.

Reviewer #2 (Remarks to the Author):

The paper present a de novo assembly of *Oryza rufipogon*, an ancestral progenitor of cultivated rice with SMRT and Hi-C data. The authors realize a multi-species comparative analysis in order to compare this progenitor with *O.sativa* and its another progenitor *O. nivara*. Comparative genomic analyses show important mutations who may affect agronomically significant traits. This new genome is the first chromosome-based of rice.

I found the paper well written and very well described in terms of methodology. I congratulate the authors for the improvements. I am convinced that the methodological choices are well thought out which was not the case in the first version. The haplotypes resolution greatly improve the manuscript. I found very easy to inspect methodology.

The authors answer to all my methodological doubts in a very detailed way. It is an important paper for the field and I recommend this paper for publication.

Response: We would again highly appreciate this reviewer's encouragement and fairly valuable comments in the last review report.

Reviewer #4 (Remarks to the Author):

The manuscript from Li et al describes the sequencing and some genomic comparisons of the *Oryza rufipogon* genome. This manuscript is significantly improved from the prior version. My main remaining critique (which is highlighted in a couple of comments below, but which applies to the entirety of the manuscript in a variety of places, is that the authors often couch their language in the context of investigating "adaptation" and "diversification", etc...However, the authors have done a lot of work to identify changes and differences across these genomes but simply identifying the changes doesn't necessarily mean that they are involved in adaptation or are causative of diversification of phenotypes. The authors should rephrase the manuscript throughout to deemphasize adaptation and diversification and to simply focus on identifying changes across genomes without having to attribute these to any evolutionary process.

Response: We would highly appreciate this reviewer's encouragement and very valuable comments. We totally agree with you and thus we rephrased the manuscript throughout to deemphasize adaptation and diversification and to simply focus on identifying changes across genomes without having to attribute these to evolutionary process. We deeply believe that we have reasonably explain the obtained results in the revised manuscript.

L26: would change the word "massive" here to "many"

Response: We would highly appreciate valuable suggestions. Done!

L28: the authors don't really follow up on large effect mutations here, so best to just say that many agriculturally relevant traits could be affected?

Response: We would highly appreciate valuable suggestions. Done!

L32: better as "genes with signatures of positive selection"

Response: We would highly appreciate valuable suggestions. Done!

L36: delete "powerfully"

Response: We would highly appreciate valuable suggestions. Done!

L54: delete "alien"

Response: We would highly appreciate valuable suggestions. Done!

L60: delete "extremely"

Response: We would highly appreciate valuable suggestions. Done!

L214: you don't really investigate adaptation here, I'd change the title to reflect the analyses better

Response: We would highly appreciate valuable suggestions. Done!

L243-248: can delete these, it's speculation that doesn't add much to the paper

Response: We would highly appreciate valuable suggestions. Done!

L249-250: you don't have the data to link these changes to phenotypic diversification. Would be best to just say you're investigating expansion and contraction of rice families

Response: We would highly appreciate valuable suggestions. Done!

L300: you can't really read anything into overall numbers of ~0.5 for dN/dS . There's a lot that can go into that. In any event, it doesn't scream purifying selection

Response: We would highly appreciate the valuable suggestion. We totally agree with you that there may be a number of factors to affect dN/dS value and infer purifying selection. To my knowledge, similar analyses in many other plant genomes suggest that almost all of them are generally under purifying selection with $dN/dS < 1$. Thus, we think that such an interpretation may be fine here.

L345: again, you don't really investigate adaptation. You identify changes, but it's hard to link these to any given adaptive event

Response: We would highly appreciate valuable suggestions. Done!